# Arbuscular Mycorrhizal Fungi Alter Arsenic Translocation Characteristics of *Iris tectorum* Maxim.

**DOI:** 10.3390/jof9100998

**Published:** 2023-10-08

**Authors:** Shuping Xing, Kangxu Zhang, Zhipeng Hao, Xin Zhang, Baodong Chen

**Affiliations:** 1State Key Laboratory of Urban and Regional Ecology, Research Center for Eco-Environmental Sciences, Chinese Academy of Sciences, Beijing 100085, China; stellar_xsp@sina.com (S.X.); zhangkangxu0129@126.com (K.Z.); zphao@rcees.ac.cn (Z.H.); bdchen@rcees.ac.cn (B.C.); 2University of Chinese Academy of Sciences, Beijing 100049, China

**Keywords:** wetland plants, arbuscular mycorrhiza, *Iris tectorum* Maxim., uptake kinetics, As species, As efflux

## Abstract

Arsenic (As) pollution in wetlands, mainly as As(III) and As(V), has threatened wetland plant growth. It has been well documented that arbuscular mycorrhizal (AM) fungi can alleviate As stress in terrestrial plants. However, whether AM fungi can protect natural wetland plants from As stress remains largely unknown. Therefore, three hydroponic experiments were conducted in which *Iris tectorum* Maxim. (*I. tectorum*) plants were exposed to As(III) or As(V) stresses, to investigate the effects of mycorrhizal inoculation on As uptake, efflux, and accumulation. The results suggested that short-term kinetics of As influx in *I. tectorum* followed the Michaelis–Menten function. Mycorrhizal inoculation decreased the maximum uptake rate (*V*_max_) and Michaelis constant (*K*_m_) of plants for As(III) influx, while yielding no significant difference in As(V) influx. Generally, mycorrhizal plants released more As into environments after 72 h efflux, especially under As(V) exposure. Moreover, mycorrhizal plants exhibited potential higher As accumulation capacity, probably due to more active As reduction, which was one of the mechanisms through which AM fungi mitigate As phytotoxicity. Our study has revealed the role of aerobic microorganism AM fungi in regulating As translocation in wetland plants and supports the involvement of AM fungi in alleviating plant As stress in anaerobic wetlands.

## 1. Introduction

Wetlands ecosystems, occupying 6% of the earth’s land surface, have high biodiversity and provide precious ecological services [1,2]. However, wetlands in many countries and regions, such as Bangladesh, India, Vietnam, China, and South America [3,4], are confronting the threat of arsenic (As) contamination, some of which have been recognized as sinks for As [1]. The widespread presence of As in wetlands mainly stems from natural processes such as rock weathering, as well as anthropogenic activities including oil exploration, insecticide and fertilizer application, and industrial sewage irrigation [5,6]. Phytotoxic As can lead to growth inhibition, physiological disorders, and ultimate death in wetland plants [7,8]. Consequently, it is necessary to understand the characteristics of As uptake and metabolism in natural wetland plants, which hold great value in safeguarding wetland plants and maintaining the ecological balance in As-contaminated wetlands [9].

Arsenic exists in various chemical forms in the environment, including inorganic species like arsenite (As(III)) and arsenate (As(V)), as well as organic species such as monomethylarsonic acid (MMA), dimethylarsinic acid (DMA), trimethylarsine (TMA), arsenocholine (AsC), arsenobetaine (AsB), and arsenosugars [10]. In natural wetlands, the primary forms of As are inorganic As(III) and As(V), which undergo dynamic interconversion due to fluctuating water table and redox conditions [11]. Moreover, distinct As species exhibit differences in the mobility, bioavailability, and phytotoxicity to plants [12]. As(III), as the dominant species under anaerobic conditions [13], is mainly translocated through aquaglycerinporins of plants [14], while As(V), which is less toxic than As(III) and prevails under aerobic conditions, is absorbed via phosphate transporters because of its chemical similarities to P [15]. Upon uptake, As(V) can be rapidly reduced to As(III) via arsenate reductase [16]. Then, As(III) may form complexes with thiols compounds such as phytochelatins (PCs), be sequestrated in vacuoles, or be pumped out from the roots, which contributes to As detoxification in plants [17]. 

Arbuscular mycorrhizal (AM) fungi, as a widespread symbiotic microorganism, play crucial roles in influencing the influx, efflux, and accumulation of As in various terrestrial plants [18,19]. The effect of mycorrhizal inoculation on plant As uptake was influenced by different AM fungi species and host plant types, for example, *Glomus mosseae* inhibited high-affinity phosphate transport and then reduced arsenate uptake in *Holcus lanatus* under short-term As exposure [20]. Conversely, in soybean inoculated with *Rhizophagus intraradices*, plant As uptake was promoted by increasing As concentrations, together with upregulated phosphate transporter *Gm*PT4 and *Ri*PT [21]. Meanwhile, mycorrhizal inoculation can increase As efflux, which has been reported in some plants including soybean [21] and maize [22]. Moreover, Sun et al. suggested that the transcripts of *Pv*Pht1;6, with great As(V) transport capacity, were strongly induced in the roots of mycorrhizal *Pteris vittata*, which finally resulted in higher As accumulation in host plants [23]. Overall, compared to terrestrial plants with numerous studies, relatively little is known about the effect of AM fungi on As translocation in wetland plants, except the artificially cultivated wetlands plant, rice [24,25]. It was demonstrated that the short-term uptake kinetics of As(III), As(V), DMA, and MMA in mycorrhizal rice [26,27] differed among rice cultivars and AM fungi species [28,29]. Mycorrhizal associations were also found to be effective in pumping out As to mitigate As stress and certain aquaporin proteins like *Os*NIP2;1 and *Os*PIP2;4 involved in As(III) efflux have been identified in rice [30]. Additionally, in the roots of upland rice, inoculation with *Funneliformis geosporum* decreased the accumulation of both As(V) and MMA [31], while inoculation with *Funneliformis mosseae* increased As accumulation. However, to date, there has been no report on the role of AM fungi in regulating As translocation in natural wetland plants.

Therefore, a common, aesthetic, and perennial natural wetlands plant, *Iris tectorum* Maxim. (*I. tectorum*), with potential capability in metal removal [32,33], yet the As translocation characteristics of which have never been studied before, was selected for experiments. Moreover, our preliminary experiment has revealed that *I. tectorum* could establish a good symbiotic relationship with AM fungi. The objectives of this investigation were to explore the influence of mycorrhizal inoculation on the short-term As uptake kinetics, As efflux, and time-dependent As accumulation in *I. tectorum*, which was exposed to solutions containing As(III) or As(V). This study can offer valuable implications for the potential application of AM fungi in alleviating As stress and enhancing the adaptability of natural wetland plants in As-contaminated wetlands.

## 2. Materials and Methods

### 2.1. Cultivation of Iris tectorum Maxim.

The seeds of *I. tectorum* were subjected to surface sterilization using 75% ethanol for 2 min and 10% sodium hypochlorite (NaClO) for 30 min, followed by soaking in sterile water. Subsequently, seeds were cultivated in a mixture of vermiculate, perlite, and coconut coir. To obtain pre-inoculated and non-inoculated plants for hydroponic experiments, uniform seedlings were then transplanted into pots with sterile mixed soil and sand (1 kg, V_soil_:V_sand_ = 1:1), with or without 50 g AM fungus inoculum per pot, and cultivated for one month in a greenhouse. The greenhouse was maintained under a relative humidity of 70%, a light intensity of 700 μmol·m^−2^·s^−1^, a light/dark cycle (16 h/8 h), and a temperature of 25 °C/18 °C (light/dark).

The AM fungus inoculum *Rhizophagus irregularis* BGC BJ09 (*R. irregularis*), which was separated from soils cultivating tomatoes in Laiguangying, Chaoyang district, Beijing, China, was subsequently propagated by a pot experiment involving *Trifolium repens* L. and a soil-vermiculite-sand mixture, which yielded approximately 56 spores/g inoculum.

### 2.2. Mycorrhizal Colonization Analysis of Pre-Inoculated Seedlings

To observe the uniformity of mycorrhizal colonization status, pre-inoculated seedlings were selected for analysis. Fresh roots were cleared in 10% KOH at 90 °C, acidified with 2% HCl, stained with 0.05% trypan blue at 90 °C, and then destained in a lactic acid-glycerol solution following the method of Phillips and Hayman [34]. For each treatment, selected root fragments were observed using a microscope (CX21FS1, OLYMPUS, Tokyo, Japan) to calculate mycorrhizal colonization indexes.

### 2.3. Short-Term As Uptake Kinetics of Iris tectorum Maxim.

Uniform seedlings of both inoculated and non-inoculated *I. tectorum* were carefully placed in 240 mL glass plant tissue culture containers, each containing 200 mL of As(III) (NaAsO_2_) or As(V) (Na_3_AsO_4_) at concentrations of 0 μM, 30 μM, 60 μM, 120 μM, 240 μM, and 480 μM. The culture solutions (pH = 5) also contained 5.0 mM MES (2-(N-morpholin)ethansulfonic acid) and 0.5 mM Ca(NO_3_)_2_. Each treatment was conducted with four replicates. After 30 min of exposure, the samples were collected, rinsed, and incubated in the ice-cold phosphate buffer solution for 10 min to remove the adsorbed As, which contained 0.5 mM Ca(NO_3_)_2_, 5 mM MES, and 1 mM K_2_HPO_4_. Finally, shoots and roots were collected for further analysis of As concentrations.

### 2.4. As Efflux from Iris tectorum Maxim.

Six inoculated and six non-inoculated *I. tectorum* seedlings were subjected to 200 mL nutrient solution containing 30 μM As(III) or As(V), with a total of four treatments and three replicates for each treatment. The nutrient solution (pH = 6) consisted of 0.3 mM KH_2_PO_4_, 1 mM CaSO_4_, 1.6 mM MgSO_4_, 0.7 mM NaNO_3_, 0.3 mM KCl, 10 μM FeNa_2_-EDTA, 20 μM H_3_BO_3_, and 7.67 μM Na_2_MoO_4_. After a 66 h exposure, all seedlings were thoroughly rinsed with de-ionized water and treated with the ice-cold phosphate buffer solution. These treated seedlings were then transferred to new culture containers with 30 mL of the same nutrient solution without As after 1 h, 2 h, 6 h, 12 h, 24 h, 48 h, or 72 h. The efflux solution was filtered through a 0.45 μm nylon filter and then analyzed for total As concentrations and As speciation.

### 2.5. Time-Dependent As Accumulation Potential of Iris tectorum Maxim.

The inoculated and non-inoculated *I. tectorum* seedlings were transferred to the containers filled with incubation solutions containing 30 μM As(III) or As(V), 5.0 mM MES, and 0.5 mM Ca(NO_3_)_2_, with a pH of 5. Each treatment was replicated four times. Sampling was conducted at intervals of 30 min, 1 h, 2 h, 6 h, 12 h, 24 h, and 48 h following exposure to the solutions. After sampling, these seedlings were thoroughly rinsed, and As concentrations were analyzed. Additionally, the As species in the plants exposed for 48 h were measured to observe possible As transformation in plants after short-term As exposure.

### 2.6. As Concentrations and As Speciation Measurement

The harvested shoots and roots were freeze-dried at −40 °C for 72 h and ground in liquid nitrogen. Approximately 0.2 g of shoots and roots were digested in 10 mL nitric acid using a microwave accelerated reaction system (Mars 5, CEM Co., Ltd., Matthews, NC, USA) [35], and then analyzed for As concentrations by inductively coupled plasma-mass spectrometry (ICP-MS, Agilent 7500, Agilent Technology, Santa Clara, CA, USA), and P concentrations by inductively coupled plasma-optical emission spectroscopy (ICP-OES, Prodigy, Teledyne Leeman, Hudson, NH, USA). Quality control was performed using a standard substance (GBW 07603, GSV-2).

Approximately 0.1 g of freeze-dried plant shoots and roots were extracted with 10 mL of 1% nitric acid using the same microwave-accelerated reaction system [18] to measure As speciation. The extracts were analyzed for As(III), As(V), MMA, and DMA, through high-performance liquid chromatography–inductively coupled plasma–mass spectrometry (HPLC-ICP-MS) (SHIMADZU 2030, Osaka, Japan). The mobile phase (10 mM (NH_4_)_2_HPO_4_ and NH_4_NO_3_, pH = 6.2) was pumped through chromatographic columns at a flow rate of 1.0 mL/min [36], with a pre column (11.2 mm, 12–20 mm) and an anion exchange column (PRP-X100, Hamilton Company, Inc., Reno, NC, USA). 

### 2.7. Statistical Analysis

The short-term concentration-dependent As uptake kinetics of *I. tectorum* was described using the Michaelis–Menten equation [28]:$$V=\frac{V_{\max}[c]}{K_{m}+[c]}$$
where *V* refers to the As uptake rate, *V*_max_ is the maximal uptake rate, *K*_m_ is the Michaelis–Menten constant equal to the substrate concentration with half the maximal transport rate, and [*c*] represents As concentrations in the nutrient solution. 

Mycorrhizal colonization was examined by a *t*-test, and kinetic parameters, the amounts of As efflux, and As concentrations in plants were tested by ANOVA. Duncan’s multiple-range tests were used to separate treatment differences (*p* < 0.05). Data analysis was completed in SPSS. All curve fitting and figure drawing were performed using Origin 2021 software.

## 3. Results

### 3.1. Mycorrhizal Colonization Status between R. irregularis and I. tectorum

No root colonization was observed in the non-inoculated *I. tectorum* plants (Table 1). The average mycorrhizal colonization rates (M%), frequencies of mycorrhiza in the root system (F%), intensities of mycorrhizal colonization (m%), arbuscule abundances in colonized root fragments (a%), and arbuscule abundances in the entire root system (A%) of all pre-inoculated plants demonstrated no significant difference between As(III) and As(V) exposure treatments (Table 1).

### 3.2. Short-Term Concentration-Dependent As Influx Kinetics in I. tectorum

The As influx into *I. tectorum* exhibited a hyperbolic rise with increasing external As concentrations, irrespective of mycorrhizal inoculation and As exposure species (Figure 1). Meanwhile, the short-term As uptake kinetics of inoculated or non-inoculated *I. tectorum* exposed to both As(III) and As(V) can be well fitted using the Michaelis–Menten equation (Figure 1 and Table 2).

Under 0–480 μM As(III) exposure, the *V*_max_ for As uptake in shoots and roots of inoculated *I. tectorum* were approximately 5-fold and 11-fold lower than that of non-inoculated plants, respectively (Table 2), and the *K*_m_ for As uptake in both shoots and roots of inoculated *I. tectorum* were 8-fold and 21-fold lower than that of non-inoculated plants, respectively (Table 2). Moreover, under 0–480 μM As(V) solutions, there was little difference between the *V*_max_ for As uptake in shoots of inoculated and non-inoculated *I. tectorum* (Table 2). However, in roots, the *V*_max_ for As uptake in inoculated plants was approximately twice as high as that in non-inoculated plants (Table 2), and the *K*_m_ values for As uptake in either shoots or roots were similar between inoculated and non-inoculated plants. Additionally, compared to that in plant roots exposed to As(III), both *V*_max_ and *K*_m_ for As uptake significantly decreased in plant roots exposed to As(V), irrespective of mycorrhizal inoculation (Table 2, *p* < 0.05).

### 3.3. Arsenic Efflux from I. tectorum

Figure 2 illustrates the amount of As efflux from *I. tectorum* exposed to As(III) (Figure 2a) or As(V) solutions (Figure 2b). Generally, both As(III) and As(V) were observed in the efflux solutions, and As(V) was the ultimate dominant species in all treatments (Figure 2). Furthermore, *I. tectorum* preloaded with As(III) solutions (Figure 2a) released a significantly higher total amount of As(III) or As(V) compared to plants preloaded with As(V) solutions in 72 h (Figure 2b, *p* < 0.01), regardless of mycorrhizal inoculation.

For plants under As(III) exposure, the cumulative amount of As(III) efflux from *I. tectorum* over time fit well to the Michaelis–Menten function (Figure 2a, *R*^2^ = 0.94 for non-inoculated plants, *R*^2^ = 0.96 for inoculated plants), regardless of mycorrhizal inoculation. Meanwhile, the cumulative amount of As(V) efflux from non-inoculated plants over time was also well described by the Michaelis–Menten function (Figure 2a, *R*^2^ = 0.99), but in inoculated plants, the cumulative amount of As(V) efflux over time fit better to the logistic function (Figure 2a, *R*^2^ = 0.96). The interaction between time and As efflux species was significant (Figure 2a, *p* < 0.001), potentially indicating a considerable difference between As(III) and As(V) efflux patterns. In general, As(III) efflux showed an initial increase followed by gradual stabilization, while As(V) efflux exhibited a continuously increasing trend (Figure 2a). Additionally, significant differences were also found in the effect of mycorrhizal inoculation on As(III) efflux and As(V) efflux (Figure 2a, *p* < 0.001). More precisely, throughout the efflux period, mycorrhizal plants had slightly higher As(III) efflux compared to non-mycorrhizal plants, yet with no significant difference (Figure 2a). In contrast, regarding As(V) efflux, mycorrhizal plants released significantly less As(V) compared to non-mycorrhizal plants in 2–24 h (Figure 2a, *p* < 0.05). However, at 48 and 72 h, the amounts of As(V) efflux from mycorrhizal plants increased and demonstrated no significant difference from that of non-mycorrhizal plants (Figure 2a).

Similar to plants under As(III) exposure, As(III) efflux and As(V) efflux from plants under As(V) exposure were also significantly different, such as in efflux rates, during the period (Figure 2b, *p* < 0.01). However, compared to plants under As(III) exposure, the cumulative amount of As(III) and As(V) efflux from *I. tectorum* all fit well to the Michaelis–Menten equation, which gradually reached a plateau (Figure 2b). Furthermore, mycorrhizal inoculation significantly influenced As(III) and As(V) efflux in plants preloaded with As(V) solutions (Figure 2b, *p* < 0.001), resulting in higher amounts of As(III) or As(V) efflux in mycorrhizal *I. tectorum* compared to non-mycorrhizal *I. tectorum*. 

### 3.4. Time-Dependent Kinetics of As and As Speciation in I. tectorum

In the As(III) exposure treatment, the total As accumulation in shoots increased linearly with time (Figure 3a, *R*^2^ = 0.81 for non-inoculated plants, *R*^2^ = 0.88 for inoculated plants, *p* < 0.001), regardless of mycorrhizal inoculation. Moreover, As concentrations in shoots of inoculated *I. tectorum* at 48 h were considerably higher than that of non-inoculated *I. tectorum* (Figure 3a, *p* < 0.05). In comparison, for roots, the total As accumulation in mycorrhizal plants also increased linearly over time (Figure 3b, *R*^2^ = 0.86, *p* < 0.001), but that in non-mycorrhizal plants followed a logistic pattern (Figure 3b, *R*^2^ > 0.99, *p* < 0.001). Furthermore, within the initial 0–24 h, mycorrhizal plant roots accumulated less As compared to non-inoculated plant roots, especially at 12 h (Figure 3b, *p* < 0.05), but at 48 h, As concentrations in mycorrhizal plant roots resulted in being significantly higher than that in non-inoculated plant roots and demonstrated a potentially continuous increasing trend (Figure 3b, *p* < 0.05). 

However, in the As(V) exposure treatment, As accumulation pattern for plant shoots showed a good fit with the logistic equation (Figure 3a, *R*^2^ = 0.96 for non-inoculated treatments, *R*^2^ = 0.97 for inoculated treatments). Furthermore, shoot As concentrations in both inoculated and non-inoculated plants significantly increased at 48 h (Figure 3a, *p* < 0.05). Moreover, during the whole As exposure period, shoot As concentrations in inoculated plants were consistently lower than that in non-inoculated plants, particularly at 1 h (Figure 3a, *p* < 0.05). Regarding root As accumulation, the curves of As concentrations exhibited a better fit to quadratic function (Figure 3b, *R*^2^ = 0.69 for non-inoculated treatments, *R*^2^ = 0.74 for inoculated treatments), irrespective of mycorrhizal inoculation. Specifically, the maximum As concentrations in the roots of inoculated plants and non-inoculated plants were observed at 12 h and 6 h, respectively, and were significantly higher than the As concentrations at respective previous time points (Figure 3b, *p* < 0.05). Moreover, As concentrations at 48 h were significantly lower than maximum As concentrations (Figure 3b, *p* < 0.05), regardless of mycorrhizal inoculation. Unlike shoots, As concentrations in inoculated plant roots were significantly lower than that in non-inoculated plant roots after the initial 2 h (Figure 3b, *p* < 0.05), but became significantly higher than that after 12 h (Figure 3b, *p* < 0.05).

Overall, at 48 h, As concentrations in plants exposed to As(III) and As(V) solutions showed no significant differences (Figure 3). Additionally, As(III) dominated in both shoots and roots (Figure 4), regardless of As exposure species and mycorrhizal inoculation. Moreover, under As(III) exposure, mycorrhizal inoculation significantly increased As(III) concentrations in shoots of *I. tectorum* (Figure 4, *p* < 0.05), but under As(V) exposure, mycorrhizal inoculation exhibited no considerable effect on As(III) concentrations in shoots and roots (Figure 4).

## 4. Discussion

Protecting plant growth in As-contaminated wetlands is crucial for maintaining ecosystem diversity and balance [37]. Numerous studies have demonstrated that arbuscular mycorrhizal (AM) fungi play a significant role in alleviating As stress in terrestrial plants [38]. However, the ecological functions of AM fungi in anaerobic wetland environments and their symbiotic effectiveness in influencing As translocation of natural wetland plants have received limited attention. This study aimed to bridge this knowledge gap through investigations into the natural wetland plant *Iris tectorum* Maxim. inoculated with *Rhizophagus irregularis*. It was demonstrated that the mycorrhizal association influences the processes of As uptake, efflux, and accumulation in the wetland plant. Furthermore, the mycorrhizal effect also varied depending on As species (As(III) or As(V)) that the AM fungi-wetland plants symbiont was exposed to.

The short-time influx kinetics for As(III) and As(V) in *I. tectorum* can be well described by the Michaelis–Menten function (Figure 1, Table 2), regardless of mycorrhizal inoculation, indicating that As(III) and As(V) uptake in inoculated or non-inoculated *I. tectorum* were both an active process relying on the binding sites and energy supply [29], similar to maize [39] and rice [29]. Concerning the kinetic parameters, the *V*_max_ for As(III) uptake in non-inoculated *I. tectorum* roots (Table 2) was lower than that of upland rice Zhonghan221 [31] and lowland rice Guangyinzhan [29], indicating that this natural wetland plant *I. tectorum* probably had weaker As uptake capabilities than artificially cultured wetland plant rice. Moreover, AM symbiosis reduced the *V*_max_ and *K*_m_ values for As(III) uptake (Table 2), suggesting a higher affinity but lower uptake rates of As(III) transporters in mycorrhizal plants, which may be one of the mechanisms through which AM fungi mitigated As(III) phytotoxicity in *I. tectorum*. Furthermore, compared to As(III) uptake, *I. tectorum* exhibited lower uptake rates for As(V), according to the decreased *V*_max_ for As(V) uptake in all treatments (Table 2). These reduced uptake rates were consistent with prior investigations reporting the 10-fold higher uptake rates of As(III) than As(V) in Aman rice var.BR11 [28]. Additionally, under As(V) exposure, mycorrhizal inoculation promoted As uptake in roots, with one-fold increased *V*_max_ compared to non-mycorrhizal plants (Table 2), which may be explained by the facilitation of As(V) uptake through induced mycorrhizal-associated phosphate transporters [23,40]. 

Apart from regulating As uptake, the efflux of As from plants into the environment is also one of the crucial mechanisms for plants to alleviate As stress [40]. It was revealed that *Solanum lycopersicum* absorbed As and released either As(III) or As(V) into the external medium [41]. Moreover, specific aquaporins, such as Lsi1 identified in rice roots, have been recognized for their important roles in mediating As(III) efflux [42], and the genes *PvACR3*, *PvACR3;2,* and *PvACR3;3*, expressed on the plasma membrane of *Pteris vittata*, were also identified as mediators of As(III) efflux and translocation [30,43]. Similarly, here, both As(III) and As(V) were detected in the collected efflux solutions, regardless of As exposure species and mycorrhizal inoculation (Figure 2). As efflux from *I. tectorum* displayed a curved increase over time, followed by a gradual slowdown in the release rates (Figure 2), which may be attributed to the rapid re-absorption of released As from the efflux solutions [41]. Furthermore, plants exposed to As(III) exhibited more As release, and required a longer time to reach a plateau (Figure 2a), which potentially resulted in lower As accumulation in plants exposed to As(III) solutions (Figure 3), and suggested an increased demand for As efflux and detoxification due to the higher phytotoxicity of As(III) than As(V), aligning with the findings of Du et al. [44]. Meanwhile, it was worth noting that mycorrhizal inoculation promoted As efflux from *I. tectorum* (Figure 2), especially under As(V) exposure, which was probably due to the involvement of specific regulatory pathways modulated by mycorrhizal associations. It has been demonstrated that As(V) exposure stimulated the expression of an arsenite efflux pump *GiArsA*, expressed in the extra-radical hyphae of *Glomus intraradices* [40]. Spagnoletti et al. also found that mycorrhizal soybean under As stress upregulated the expression of gene, *RiArsA*, encoding a putative As efflux pump in *Rhizophagus intraradices* [21]. Nonetheless, the precise molecular mechanisms governing As efflux in plants remain elusive, and elucidating the promotion effects of mycorrhizal inoculation on As efflux in natural wetland plants warrants further investigation at the molecular level.

Considering co-existing As influx and efflux processes (Figure 1 and Figure 2), time-dependent As accumulation dynamics were observed in the shoots and roots of plants exposed to 30 μM As(III) or As(V) solutions for 48 h (Figure 3). In this study, under As(III) exposure, roots of non-inoculated *I. tectorum* displayed a progressively stabilized trend in As accumulation (Figure 3b), while the shoots and roots of inoculated *I. tectorum*, together with shoots of non-inoculated *I. tectorum* (Figure 3), all demonstrated linear increments of As concentrations over time. Moreover, within the initial 24 h for As(III) exposure, As concentrations in mycorrhizal plant shoots and roots were lower than that in nonmycorrhizal plant shoots and roots (Figure 3), which may be attributed to lower *V*_max_ of mycorrhizal plant shoots and roots (Table 2). However, after As(III) exposure for 48 h, mycorrhizal inoculation significantly increased As concentrations in both shoots and roots (Figure 3, *p* < 0.05). This can be ascribed to, firstly, promoted As(V) reduction into As(III) (Figure 4), which supported subsequent As detoxification pathways including As(III) complexation with thiols, As(III) sequestration in vacuoles, and As(III) efflux, and led to higher As concentrations in mycorrhizal *I. tectorum* [17,38]. Secondly, some mycorrhizal structures, such as arbuscules and hyphae, contributed to As sequestration and enhanced As accumulation in host plants [45]. It was worth mentioning that, unlike plants exposed to As(III) solutions, in *I. tectorum* exposed to As(V) solutions (Figure 3), As concentrations dynamics in plant roots fitted well with a quadratic function (Figure 3b). The presence of declining As concentrations in inoculated and non-inoculated plant roots was observed at 24 h (Figure 3b), together with a simultaneous increase in As concentrations in plant shoots (Figure 3a), suggesting that a part of As was probably translocated from roots to shoots [17]. Furthermore, this decline in As concentrations of roots may also be ascribed to the increased As efflux in plants exposed to As(V) (Figure 2b). Furthermore, in contrast to As(III) exposure, mycorrhizal inoculation demonstrated no significant impact on ultimate As accumulation in plants exposed to As(V) at 48 h (Figure 3), which may be related to several certain factors mediating As accumulation under As(V) exposure, such as As(V) concentrations in solutions, distinctive phosphate transporters kinetics, and the competitive relationship between As(V) and P [17]. Overall, the findings revealed complex interactions between AM fungi and *I. tectorum* in translocating As under As(III) and As(V) exposure, and further comprehensive investigations are important to elucidate the mechanisms underlying mycorrhizal-mediated responses to different As species in natural wetland plants.

## 5. Conclusions

This is an original investigation into the effects of AM fungal inoculation on the As uptake, efflux, and accumulation in natural wetland plants. It revealed that AM symbiosis altered short-term As uptake characteristics of *I. tectorum* in response to As(III) and As(V) stress. Additionally, AM symbiosis promoted As efflux from plants when exposed to either As(III) or As(V), potentially mitigating As toxicity to plants. In general, mycorrhizal plants tended to accumulate more As than nonmycorrhizal plants after 48 h As(III) or As(V) exposure, and facilitate the As(V) reduction. These findings give insights into the role of AM fungi in plant growth and As homeostasis in As-contaminated wetlands and provide a theoretical foundation for the improvement of As tolerance of natural wetland plants. Further molecular research is required to unravel the mechanisms underlying As translocation in the symbiont of AM fungi-natural wetland plants.

## Figures and Tables

**Figure 1 jof-09-00998-f001:**
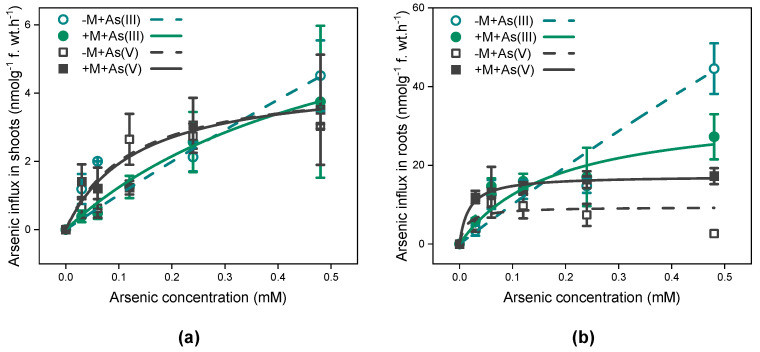
Concentration-dependent kinetics for As influxes in shoots (**a**) and roots (**b**) of *Iris tectorum* Maxim. Each point is presented as mean ± SE (*n* = 4). “−M + As(III)” and “+M + As(III)” represent non-inoculated (green hollow circle) and inoculated plants (green filled circle) exposed to solutions containing arsenite (0–480 μM), respectively. “−M + As(V)” and “+M + As(V)” represent non-inoculated (black hollow square) and inoculated plants (black filled square) exposed to solutions containing arsenate (0–480 μM), respectively.

**Figure 2 jof-09-00998-f002:**
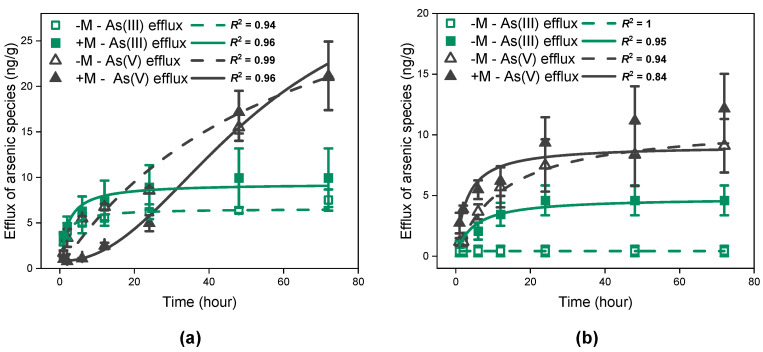
As(III) and As(V) efflux of *Iris tectorum* Maxim. exposed to 30 μM arsenite (**a**) or arsenate (**b**) for 72 h. Each point is represented as mean ± SE (*n* = 3). “−M − As(III) efflux” and “+M − As(III) efflux” represent As(III) released by non-inoculated (green hollow square) and inoculated plants (green filled square). “−M − As(V) efflux” and “+M − As(V) efflux” represent As(V) released by non-inoculated (black hollow triangle) and inoculated plants (black filled triangle).

**Figure 3 jof-09-00998-f003:**
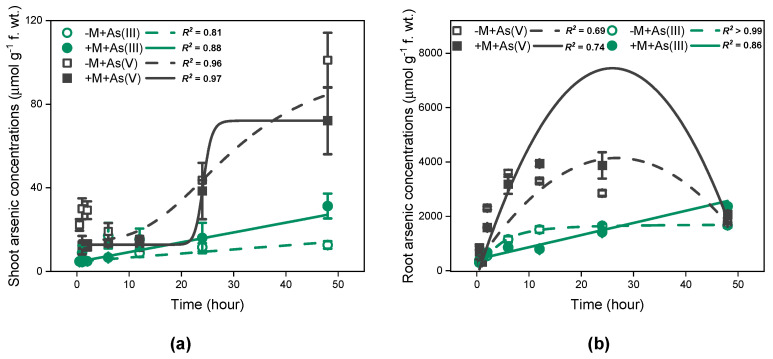
Time-dependent kinetics for As accumulation in *Iris tectorum* Maxim. Shoots (**a**) and roots (**b**). “−M + As(III)” and “+M + As(III)” represent non-inoculated plants (green hollow circle) and inoculated plants (green filled circle) exposed to solutions containing As(III), respectively. “−M + As(V)” and “+M + As(V)” represent non-inoculated plants (black hollow square) and inoculated plants (black filled square) exposed to solutions containing As(V), respectively. Each point is the mean ± SE (*n* = 4).

**Figure 4 jof-09-00998-f004:**
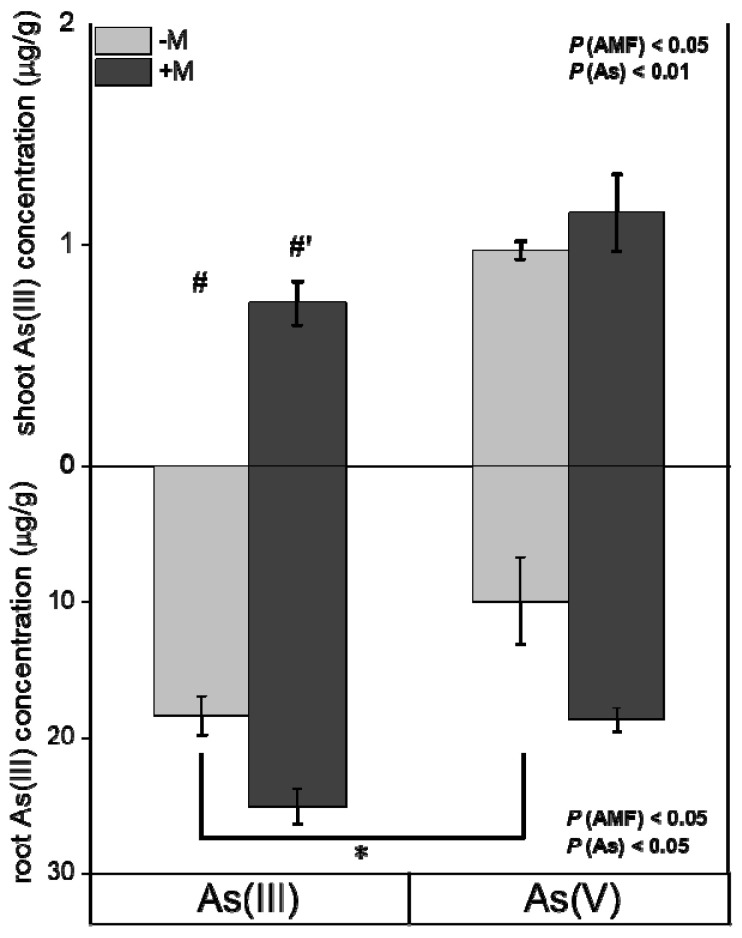
Concentrations of different As species in *Iris tectorum* Maxim. after exposure to solutions containing 30 μM As(III) or As(V) for 48 h. Data are mean ± SE (*n* = 4). “−M” and “+M” represent non-inoculated (grey bars) and inoculated (black bars) treatments, respectively. “As(III)” and “As(V)” represent treatments exposed to solutions containing As(III) and As(V), respectively. “*” indicates a significant difference between two As species treatments at *p* < 0.05 (*t*-test) in the same inoculation treatment. “#, #′” indicate a significant difference at *p* < 0.05 between inoculated and non-inoculated plants in the same As species treatment (*t*-test).

**Table 1 jof-09-00998-t001:** Mycorrhizal colonization rate of *Iris tectorum* Maxim.

	Treatments			
	As(III)		As(V)	
	−M	+M	−M	+M
F%	0	88.10 ± 2.01	0	88.55 ± 3.98
M%	0	19.94 ± 4.11	0	16.73 ± 4.79
m%	0	16.73 ± 4.79	0	12.22 ± 5.19
a%	0	50.09 ± 6.90	0	43.16 ± 3.02
A%	0	10.83 ± 3.19	0	6.20 ± 2.78

“As(III)” and “As(V)” represent treatments exposed to arsenite and arsenate solutions, respectively. “−M” represents nonmycorrhizal inoculation, and “+M” represents mycorrhizal inoculation. Data are shown as mean ± SE (*n* = 6).

**Table 2 jof-09-00998-t002:** The kinetic parameters for As influx in *Iris tectorum* Maxim.

Treatments		*V* _max_		*K* _m_		*R* ^2^	
		nmol g^−1^ Fresh Weight		mM			
		Shoot	Root	Shoot	Root	Shoot	Root
As(III)	−M	42.74 ± 180.21	395.20 ± 1275.13 *	4.09 ± 19.16	3.83 ± 13.80	0.91	0.93
	+M	8.11 ± 1.11	34.61 ± 8.73	0.56 ± 0.13	0.18 ± 0.12	0.99	0.72
As(V)	−M	4.44 ± 1.59	9.44 ± 1.75 *	0.12 ± 0.10	0.01 ± 0.02	0.77	0.24
	+M	4.50 ± 0.63	17.32 ± 0.63	0.13 ± 0.06	0.02 ± 0.00	0.90	0.90

‘*’ means significant difference in *V*_max_ for As influx of non-inoculated plants exposed to As(III) and As(V) solutions (*p* < 0.05).

## Data Availability

The datasets generated during the current study are available from the corresponding author on reasonable request.

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
