# Peer review of "Arbuscular Mycorrhizal Fungi Alter Arsenic Translocation Characteristics of Iris tectorum Maxim."

_jof, 2023, doi:10.3390/jof9100998_

Round 1
Reviewer 1 Report
The manuscript compares how Iris plants exposed to sources of As are influenced by VA mycorrhizal inoculations. The paper is of interest to the mycorrhizal and ecological community, but requires a few small clarifications:
The authors correctly state in the Introduction that wetland soils are frequently anaerobic. It appears that the experiments were conducted under more or less aerobic conditions; therefore the authors should elaborate in the Discussion if different results would have been expected, had experimental conditions been more anaerobic.
VA mycorrhiza (or arbuscular mycorrhiza) would be an important keyword to add (instead of mycorrhizal symbiosis).
Lines 28-29: ecosystems, occupying 6% of the earth’s land surface, has have high biodiversity and provides ....
Line 87: How long was surface sterilization?
Line 91: Explain what it means that seeds were "pre-inoculated".
95: Indicate the source of BJ09
101: distained destained
330-331: The authors state that "...genes .... localized on the plasma membrane... " This should be corrected/clarified. Gene products? Genes expressed?
Generally a high standard of English. Only a few minor issues were observed.
Reviewer 2 Report
This publication presents interesting results of the effect of mycorrhizal association on the processes of As uptake, efflux, and accumulation in Iris tectorum Maxim The adopted approach in this study was very interesting since AMF showed a promising asset to alleviate the deleterious effects of As stress on crops the wetland ecosystem.
The manuscript was well introduced, and the authors adopted very convincing methods with a consistent discussion of the different obtained results. However, the manuscript needs minor revisions to be suitable for publication in Journal of Fungi.
General comments
The English of this manuscript needs slight improvement.
Other comments
Abstract
1. Keywords: please add Iris tectorum.
Introduction
2. L63: please correct the citation form.
M&M
3. L90: please add a space between the value and the unit. Please check throughout the manuscript.
4. L102: please provide “author et al.” before the reference number.
5. L122: please change “30 ml the same” to “30 ml of the same”.
Results
6. Table 2: please check the SE of Vmax values recorded for As(III) in -M plants.
7. L132: please change “specie” to “species”.
8. L276: please change “increase” to “increased”.
Discussion
9. L341 and L346: please correct the citation form.
The english of the manuscript needs slight revision.
Reviewer 3 Report
Dear authors.
1)Please, explain why you have chosen Iris for studying AMF protection against As pollution in conditions of wetlants. As far as I know high phytoremediation ability is typical for Phragmites australis and Typha angustifolia which are common for wetlands. Any peculiarities of Iris tectorum Maxim????
2)As a proposal for your future work it seems highly important to study the AMF effect on selenium influx and release into environment in conditions of As loading, as Se is known to protect plants against As. In terrestrial plants AMF is known to improve Se accumulation by plants bit no information so far has been obtained as for the wetland conditions.
3)Besides, it seems of highly importance to study changes in antioxidant status of Iris treated with AMF – another way of plant protection. If you have any data on these topics, it will be fine to add the data to the text.
In any case these are only proposals and suggestions for your future work and not a drawbacks of your work
